# Structure establishment of three-dimensional (3D) cell culture printing model for bladder cancer

Myeong Joo Kim[1]☉, Byung Hoon Chi[1]☉, James J. Yoo[2], Young Min Ju[2], Young Mi Whang[1]‡*, In Ho Chang[1]‡*

**1** Department of Urology, College of Medicine, Chung-Ang University, Seoul, Republic of Korea, **2** Wake Forest Institute for Regenerative Medicine, Wake Forest School of Medicine, Medical Center Boulevard, Winston-Salem, NC, United States of America

☉ These authors contributed equally to this work.
‡ These authors also contributed equally to this work.
* caucih@cau.ac.kr (IHC); ymwhang@gmail.com (YMW)

**Data Availability Statement:** All relevant data are in the paper.

**Funding:** This research was supported by the Basic Science Research Program through the National Research Foundation of Korea (NRF) funded by the

## Abstract

### Purpose

Two-dimensional (2D) cell culture is a valuable method for cell-based research but can provide unpredictable, misleading data about *in vivo* responses. In this study, we created a three-dimensional (3D) cell culture environment to mimic tumor characteristics and cell-cell interactions to better characterize the tumor formation response to chemotherapy.

### Materials and methods

We fabricated the 3D cell culture samples using a 3D cell bio printer and the bladder cancer cell line 5637. T24 cells were used for 2D cell culture. Then, rapamycin and Bacillus Calmette-Guérin (BCG) were used to examine their cancer inhibition effects using the two bladder cancer cell lines. Cell-cell interaction was measured by measuring e-cadherin and n-cadherin secreted via the epithelial-mesenchymal transition (EMT).

### Results

We constructed a 3D cell scaffold using gelatin methacryloyl (GelMA) and compared cell survival in 3D and 2D cell cultures. 3D cell cultures showed higher cancer cell proliferation rates than 2D cell cultures, and the 3D cell culture environment showed higher cell-to-cell interactions through the secretion of E-cadherin and N-cadherin. Assessment of the effects of drugs for bladder cancer such as rapamycin and BCG showed that the effect in the 2D cell culture environment was more exaggerated than that in the 3D cell culture environment.

### Conclusions

We fabricated 3D scaffolds with bladder cancer cells using a 3D bio printer, and the 3D scaffolds were similar to bladder cancer tissue. This technique can be used to create a cancer cell-like environment for a drug screening platform.

Ministry of Education, Science, and Technology, Republic of Korea (2017R1D1A1B03031514, 2018R1D1A1A02050248), and the Korea Health Technology R&D Project (HI17C0710). The funders had no role in study design, data collection and analysis, decision to publish, or preparation of the manuscript.

**Competing interests:** The authors have declared that no competing interests exist.

## Introduction

The cell culture system was an essential method that is often used in basic and clinical *in vivo* studies. Cell culture is an important technique in the drug discovery process, providing a simple, fast, and cost-effective way to reduce animal testing.[1] Two-dimensional (2D) cell culture is a valuable method for cell-based research but has limitations.[2] Almost all cells in the *in vivo* environment are surrounded by extracellular matrix (ECM) and other cells. Hence, 2D cell culture sometimes provides unpredictable data that can be misleading regarding the *in vivo* response.[3] Currently, standard procedures for compound screening in new drug development begin with 2D cell culture-based testing and then move to animal model testing and clinical trials. Only about 10% of tested compounds are successfully processed through clinical development and many drugs fail during clinical trials.[4] However, 2D culture conditions do not faithfully reflect the situation *in vivo* since proper tissue structure and cell-to-cell interactions are lost.[5] Therefore, it is essential to develop and establish an *in vitro* cell-based system that can simulate cellular behavior *in vivo* more realistically. 3D *in vitro* tumor models have been successfully used to evaluate efficacy and tissue pharmacokinetics of anticancer drugs. 3D spheroids models have been studied to reproduce the spatial organization and microenvironmental factors of *in vivo* micro-tumors more accurately, such as relevant gradients of nutrients and other molecular agents, and It is possible to generate cell-to-cell and cell-to-matrix interactions by them. [6] Although more advanced compared to two-dimensional culture, 3D spheroid models lack major ECM elements of the tumor microenvironment. To overcome this, 3D bioprinting techniques with scaffold bioink made up of cellular material and additives such as growth factors, signaling molecules, etc. have been utilized. Compared to traditional tissue engineering methods, the technologies utilized by 3D bioprinting systems allow for greater precision in the spatial relationship between the individual elements of the desired tissue. As advances of computer aided design (CAD), 3D bioprinting offers great potential for regenerative medicine applications. We focused on the effects of the rapamycin mammalian target (mTOR) pathway and Bacillus Calmette-Guérin (BCG). The mTOR pathway is the most commonly mutated signaling pathway in many cancers, and BCG is currently the drug of choice for bladder cancer treatment.[7] The loss of pathway inhibition is generally associated with a variety of cancers that results in unrestrained activation of the PI3K pathway, leading to less control of cancer cell proliferation.[8] BCG is among the most effective immune therapeutics for non-muscle-invasive bladder cancer patients and has been used for more than 30 years.[9, 10] An inhibitor that regulates the mTOR pathway activity was used[11], and the antitumor effect of BCG was confirmed. Rapamycin and BCG are effective in the 2D cell culture model but have no effect in patients. In particular, rapamycin use is limited in clinical studies[12]. Approximately 30 to 50% of patients undergoing BCG therapy do not respond within the first 5 years of treatment, and its use is limited because of side effects[13]. The development of an appropriate three-dimensional (3D) cell culture model system could better simulate the cancer micro-environment. We hypothesized that the effect of rapamycin (mTOR inhibitor) and BCG in the 3D cell culture system would be less than that observed in 2D, indicating that 3D cell culture is a more suitable *in vivo* model.

## Materials and methods

### Cells and reagents

The human bladder cancer 5637 and T24 cell lines were purchased from the American Type culture collection (Manassas, VA, USA). 5637 and T24 cells were maintained on RPMI 1640 medium supplemented with 10% fetal bovine serum and 100× penicillin/streptomycin (Gibco,

MD, USA) in a humidified incubator with 5% $CO_2$ at 37˚C. Rapamycin (inhibitor of mTOR) was purchased from Sigma-Aldrich (St, Louis, MO, USA). *Mycobacterium bovis* BCG was obtained as a commercial lyophilized preparation (Onco-Tice, NJ, USA). BCG was resuspended in phosphate buffered saline (PBS; Hyclone, Logan, UT, USA) and aliquots with a multiplicity of infection (MOI) of 100 ($1 \times 10^7$ cells/ml) were prepared and stored at -80˚C until use. A GelMA prepolymer solution was used (Gel4Cell; Bioink Solutions, Daegu, Korea).

## Cell culture and construct fabrication

The 2D cell culture samples were seeded ($1 \times 10^6$ 5637 or T24 cells) on 60 mm plates. A 3D cell printer (In vivo; Rokit, Seoul, Korea) was used to fabricate 3D cell cultures. 5637 and T24 cells at a density of $1 \times 10^6$ cells/ml were collected by centrifuging at 1300 rpm for 3 min and suspended in GelMA polymer solution for Gel4Cell. The mixtures were gently stirred to ensure that the cells were evenly distributed, and 1 ml was drawn from the mixture into a sterilized syringe with a 25 gauge needle. The user-created branched constructs design was loaded into the computer, and the mixtures were extruded from the syringe needles into the low temperature chamber. The temperature of the nozzle fixer was maintained at 4˚C and the plate bed at 10˚C. They were controlled by a computer program design model (Creator K) by moving the nozzles in the X and Y directions and the platform in the Z direction.

## Crosslinking conditions setting

The mixture was physically crosslinked by exposure to UV light (356 nm) using a UV lamp (KA.TN-4LC). The UV exposure time for crosslinking was determined after setting the conditions to 0, 30, 60, 90, 120, and 180 s. Each construct was cultured in a 60 mm plate dish with 3 ml complete culture medium.

## Live/dead staining assay

Cell survival rate in the 3D cell constructs was assessed on days 1, 3, and 5 after biofabrication. A fluorescent live/dead staining solution (Thermofisher, MA, USA) was utilized according to the manufacturer's instructions. Each 3D cell construct and 2D culture cell sample was washed in PBS three times before staining. The mixture of Calcein-AM (2 μM) and EthD-1 (4 μM) was filtered through a 0.22-mm syringe filter (Sigma, MO, USA). The cell morphologies were observed under fluorescence microscopy (Leica DMI8). Three independent samples were observed.

## Cell proliferation assay

The Cell Counting Kit-8 (CCK-8; Dojindo, MD, USA) was used to analyze cell proliferation in 3D cell constructs and 2D-cultured cells on days 1, 2, and 3 according to the manufacturer's instructions. 3D cell constructs and 2D-cultured cells were washed with PBS three times. Then, 1 ml PBS and 0.1 ml CCK-8 solution were added to each 60-mm cell culture dish and incubated in the dark for 3 h with 5% $CO_2$ at 37˚C. After incubation, 0.2 ml of medium was transferred to a 96-well plate and immediately the fluorescence was determined at an excitation of 450 nm using a microplate reader (SpetraMax i3x, Molecular Devices). The 3D constructs and 2D petri dishes without cells were used as controls. Three independent samples were tested in each group.

## Cell viability assay

The 5637 and T24 cells were plated in 96-well plates at $5 \times 10^3$ cells per well in complete medium and treated with rapamycin (1 μM) and BCG (30 MOI) for 1 day. After 24, 48, and 72

h, cell viability was analyzed using the 3-(4,5-dimethylthiazol-2-yl)-2,5-diphenyltetrazolium bromide (MTT) assay according to the manufacturer's instructions (Sigma-Aldrich, MO, USA). The fluorescence intensity was measured with a fluorescence microplate reader (spetra-Max i3x, Molecular Devices).

## Western blot analysis

Cells were collected after 24h treatment, kept on iceand the protein concentrations were measured using a BCA Protein Assay Kit (Thermofisher, Waltham, MA). Equal amounts of protein samples (20 μg/lane) were separated SDS-PAGE gels and then transferred onto polyvinylidene difluoride (PVDF) membranes (Millipore, MA, USA). The membranes were incubated with the primary antibodies overnight at 4˚C and next day the secondary antibody (1:5000) was added, incubated for 1h at room temperature. The protein bands were detected with a Chemidoc imaging system (Bio-Rad). The following antibodies from Cell Signaling Technologies (MA, USA) were used: rabbit polyclonal antibodies against phosphorylated mTOR (Ser2448), phosphorylated p70s6k (Ser371), and phosphorylated 4E-BP1 (Ser65). The antibody to monoclonal mouse GAPDH was purchased from Santa Cruz (TX, USA). All experiments were performed in triplicate.

## ELISA

IL-6 (D6050), IL-12 (D1200), IFN-γ (DIF50), Duo-set ELISA kit, and the E-Cadherin (DCADE0) quantikine ELISA kit were purchased from R&D Systems (Minneapolis, MN, USA). We used an N-Cadherin (E-EL-H0195) ELISA kit (Elabscience, Houston, TX, USA). 2D Cells were seeded at $1 \times 10^6$ cells in a 60 mm cell culture dish. The 3D cell culture constructs of 5637 cells and T24 cells were seeded at a density of $1 \times 10^6$ cells/ml in GelMA solution. The next day, dishes were treated with TGF-β1 (5 ng/ml) and cultured for 24 h. After 48 h, the cell culture supernatant was collected from the sterilized container and centrifuged at a speed of 3000 rpm for 10 min at 4˚C. Samples were immediately analyzed or aliquoted and stored at -80˚C prior to analysis. Following the manufacturer's instructions, the OD of the wells was measured with a microplate reader (SpetraMax i3x, Molecular Devices). The microplate reader device was set to 450 nm with a wavelength of 540 nm.

## Statistical analysis

All data are presented as the mean ± standard deviation (SD) of at least three separate experiments performed in triplicate. Data were compared using Student's *t*-test. P values less than 0.05 were considered statistically significant.

# Results

## 1. 3D bio-printing process and construct patterning platform

The bio-printing process for the 3D cell structure was performed using the bladder cancer cells 5637 and T24 with GelMA. To determine the conditions for the platform, we used a $1 \times 1 \times 0.01$ mm net pattern structure with the creator K program (Fig 1).

## 2. Establishment of the conditions of UV crosslinking and size

The requirements of materials crosslinking time can be different, as the materials need to be extruded from syringe, resist deformation after printing, and maintain structural integrity over the lifetime of the printed. Constructs with low crosslinking times of 30 and 60 s gradually flowed over the vial. However, constructs with crosslinking times of 90, 120, and 180 s were

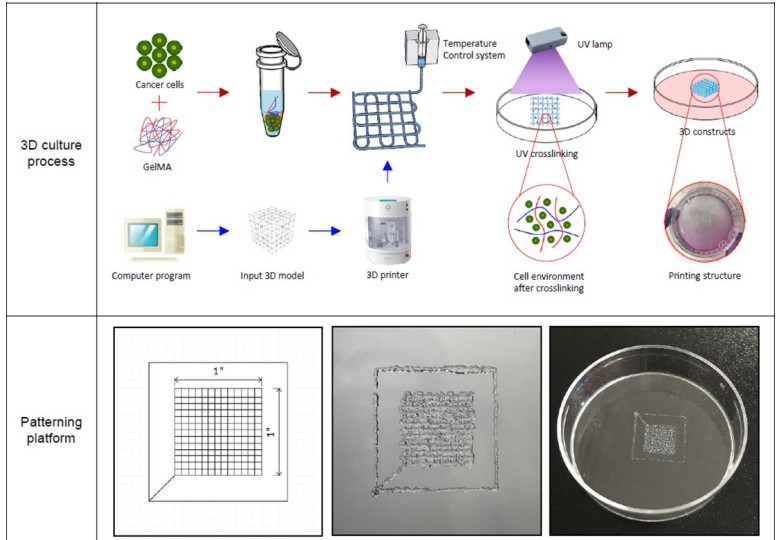

**Fig 1. Cell culture using simple to complex three-dimensional bio-printing models.** Schematic illustration of the 3D bio-printing process and image of the printing setup and constructs. Platform designed using a computer program. The overall size is $1 \times 1 \times 0.01$ mm, and the order is x, y, and z-axis (left image). Top view of 3D bio-printing construct used GelMA demonstrating the porous platform of the scaffold (middle, right images).

observed to maintain their shape (Fig 2A). Only the structures that maintained their shape at 90, 120, and 180 s after printing were crosslinked, and the others melted in the medium (Fig 2B). We measured the force of the structure at 90, 120, and 180 s at a constant time (0.7 s) and distance (1.225 mm). The intensities were 5.1428, 5,8687, and 7.8597 N, respectively, and these results showed that 180 s crosslinking makes the structures hard (Fig 2C). The 5637 cell was used to confirm the shape of the construct according to the binding time based on the conditions of the crosslinking time. At crosslinking times of 90 and 120 s, the structure was similar to the planned pattern, but at 90 s, we found that there were holes or they had melted. At 180 s, the structure become difficult to distinguish (Fig 2D). This can be controlled by stacking layers with many cells into 3D structure. Our ultimate goal is to maintain the structure of bladder cancer cells in 3D for more than 2D. So we confirmed that the shape remained after 5 days by comparing the GelMA only and cell/GelMA mixture structures (Fig 2E). There was interconnectivity of the different layers of scaffolds in the Z-direction. We printed 0.03, 0.08, 0.1, and 0.15 mm layers to determine the appropriate thickness. The 0.03 mm-layer structure was melted and the 0.1 mm- and 0.15 mm-layer structures had shape but collapsed and melted. The 0.08 mm-layer structure shape was well-maintained on the plate (Fig 2F). Therefore, the optimal conditions for our experiment were 120 s for crosslinking time and 0.08 mm for height.

## 3. Cell survival and proliferation rates in 2D and 3D culture models

The 5637 and T24 cells were plated in a 60 mm dish at a density of $1 \times 10^5$ cells. The cells were examined using live/dead staining. The number of dead cells stained in the 3D construct was lower than that in the 2D cell culture model. In addition, it was confirmed that the cell growth was better than that in the 2D cell culture (Fig 3A). We compared the proliferation of 5637 cells (% of day 1, mean ± SE) in the 2D cell culture group with the 3D cell culture group on days 2 and 3 and subtracted the control data [(4.89 ± 3.77 [2D] vs. 25.84 ± 3.33 [3D], day 2, p < 0.05), (13.39 ± 4.04 [2D] vs. 45.19 ± 3 [3D], day 3, p < 0.01)] and confirmed the results in

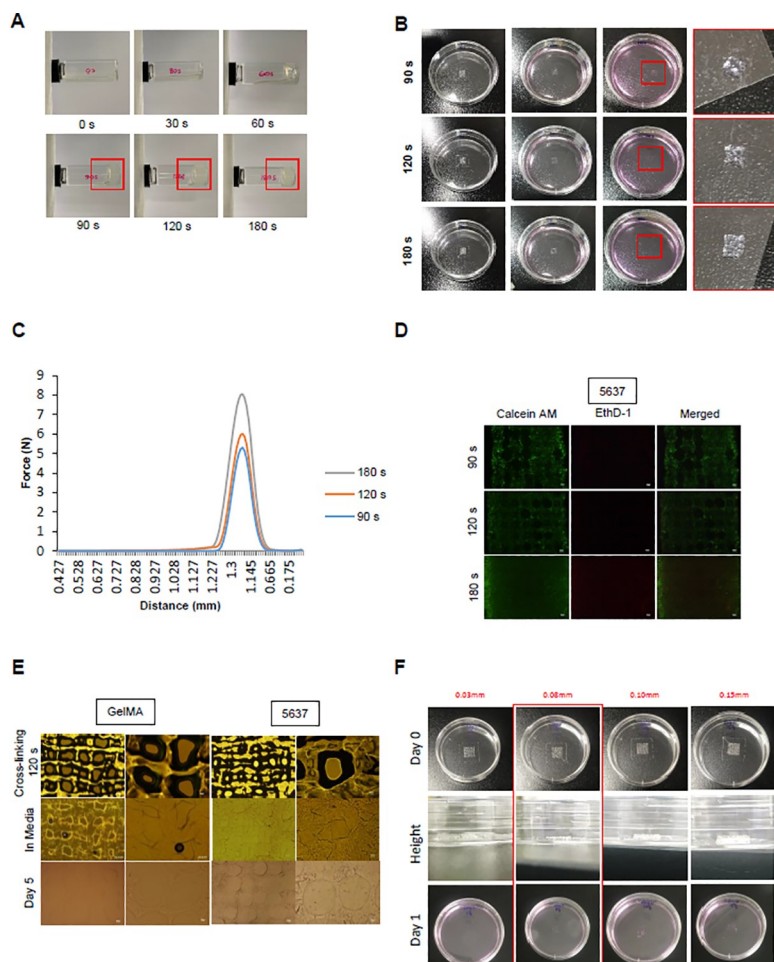

**Fig 2. UV crosslinking condition for 3D culture model.** (A) After adding 1 ml to the vials, the UV crosslinking time condition was confirmed. The times were fixed at 0, 30, 60, 90, 120, and 180 s, and a 365 nm UV lamp was used. (B) The pattern was printed on a 60-mm plate using GelMA. After crosslinking at the conditions provided in (A), complete media was added and samples were incubated for 1 day in a 37°C, 5% $CO_2$ incubator. (C) The shape of the structure was measured after 90, 120, and 180 s. The intensity force at which the structure was destroyed under the conditions of constant distance (1.225 mm) and time (0.7 s) was measured. (D) The GelMA structure and GelMA/ bladder cancer cells mixture was printed and crosslinked. After being placed in complete media, the GelMA structure was confirmed with a microscope, and the structure was maintained after incubation for 5 days. (E) To investigate the effect of crosslinking time on the cells, we performed live/dead staining. After 1 day of culture, the printed mixture of GelMA and cells was treated with Calcein-AM (2 μM) and EthD-1 (4 μM) and examined by fluorescence microscopy. The shape of the structure was examined. (F) The different layers of scaffold in the Z-direction. The constructs were printed at different heights of 0.03, 0.08, 0.1, and 0.15 mm with only GelMA. Some of the structure was maintained even after 1 day in the complete media.

the T24 cells (88.70 ± 4.83 [2D] vs. 151.38 ± 4.72 [3D], day 3, p < 0.01). The 5637 cells showed a difference in cell growth on days 2 and 3 between 3D and 2D cell culture groups, and T24 cells showed a difference on day 3. The results show that cell growth rates were better in 3D cell constructs than in 2D cell cultures (Fig 3B).

## 4. The effect of rapamycin and BCG in 2D and 3D culture models

Cells were evaluated using the CCK-8 assay. The cell viability of 5637 and T24 cells was inhibited by rapamycin and BCG treatment. However, the 2D cell culture showed a higher level of

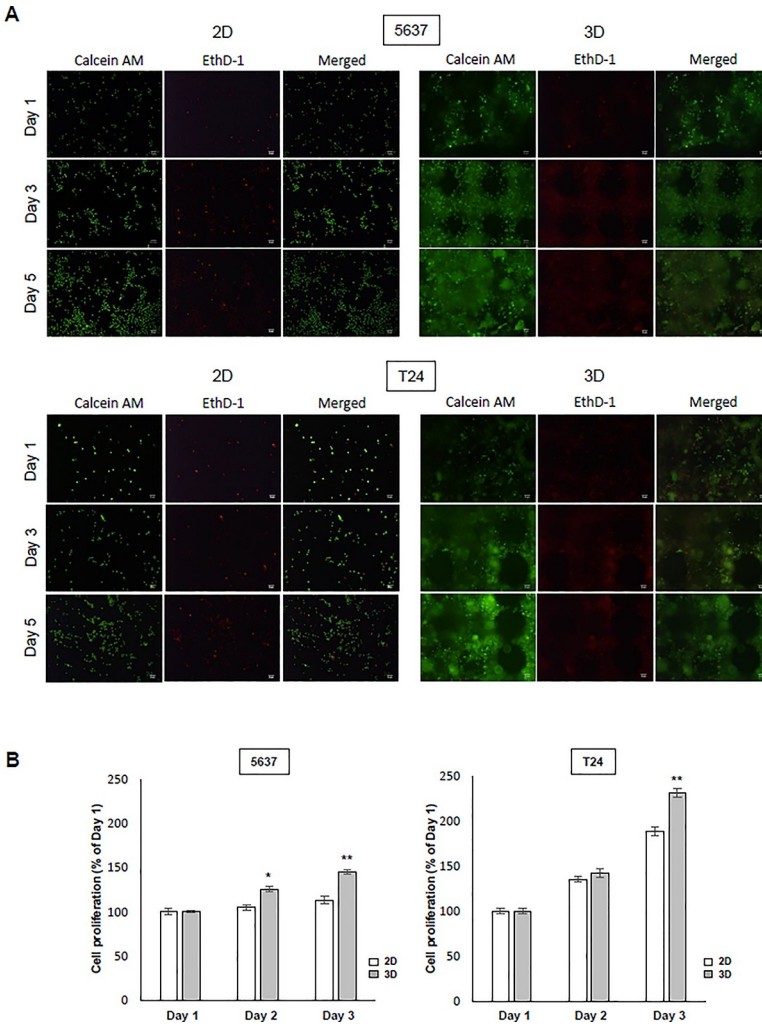

**Fig 3. Morphological changes in the cells of printed constructs and comparison of 2D and 3D cell proliferation.**
(A) To confirm the morphology of the bladder cancer cells, $1 \times 10^5$ 5637, T24 cells were seeded on a 60-mm plate dish, stained with fluorescence markers, and observed with a microscope for 5 days. For 3D cell culture, $1 \times 10^5$ cells were mixed in 1 ml solution and then the construct was printed. The cells ($1 \times 10^5$) were used in a total of 100 µl solution to print one structure. Live/dead staining in the 2D and 3D cell culture environment confirmed cell viability. Cells were stained with Calcein AM (2 µM) and EthD-1 (4 µM) and observed after 1, 3, and 5 days with a fluorescence microscope. (B) The 5637 and T24 human bladder cancer cells were seeded on 60-mm plates for 1, 2, and 3 days. Cell proliferation was determined by the CCK-8 assay in bladder cancer cells. $^* p < 0.05$, $^{**} p < 0.01$, ratios were compared between 2D and 3D cultures on each day. Data are the mean ± SEM (n = 6).

suppression than the 3D cell culture model. In the case of 5637 cells, the 2D cell culture showed a great decrease in cell viability in the control with each day in the untreated group due to rapamycin (1 µg/ml), and based on in the control, each untreated group in the 3D cell culture model was decreased. When Bacillus Calmette-Guérin (BCG) (30 MOI) was administered in each comparative control group, viability was decreased in the 2D and 3D cell culture model. T24 cells were also dramatically decreased by the rapamycin in 2D cell culture and this was also observed in the 3D cell culture model. When BCG was administered in the control group, viability decreased in the 2D cell culture environment and decreased in the 3D cell culture environment (Fig 4A). Overall, the 2D cell culture model showed dramatic results, but there was no significant difference in the 3D cell culture model. We compared 2D and 3D cell

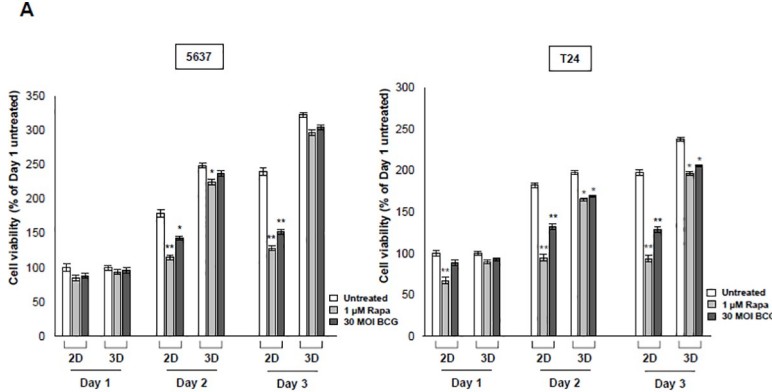

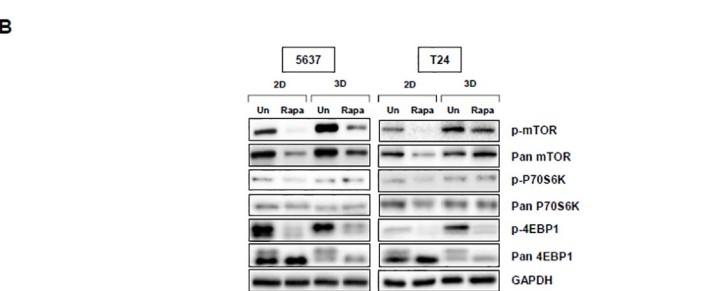

**Fig 4. Drug treatment of bladder cancer cells in 2D and 3D environments.** (A) Effect of treatment with rapamycin (inhibitor of mTOR) and Bacillus Calmette-Guérin (BCG) on viability of bladder cancer cells. Cells were treated with indicated concentrations of rapamycin (1 μM, BCG, 30 MOI) for 1, 2, and 3 days. Absorbance was measured at 540 nm and cell viability was determined by the MTT assay. * $p < 0.05$, ** $p < 0.01$, ratios were normalized to each untreated group. Data are the mean ± SEM (n = 6). (B) Western blot analysis of 5637 and T24 bladder cancer cells cultured in 2D and 3D compared the expression of mTOR-mediated p70s6K and 4E-BP1 signal pathways. Cells were treated with rapamycin (1 μM) for 24 h and then the cells were lysed. Western blotting results showed a reduced level of expression in cells in the 2D environment due to treatment with rapamycin. GAPDH was used as a loading control. A representative result from three experiments is shown.

culture model by identifying the mTOR pathway with 4E-BP1 and p70s6k to simulate the autophagic process by inhibiting with rapamycin. We tested whether rapamycin affects the status of the mTOR phosphorylation by inducing autophagy in 5637 and T24 cells. Phosphorylation of mTOR, 4E-BP1, and p70s6k was suppressed in all rapamycin-treated cells, but the 3D cell culture environment showed a decrease in phosphorylation compared with the 2D cell culture (Fig 4B). These results showed that the drugs had less of an effect in the 3D cell culture model.

## 5. The BCG effect on cytokine production in the 2D and 3D cell culture environment

We measured the levels of several cytokines (IL-6, IL-12, and IFN-γ) in the 2D and 3D bladder cancer cell models after BCG treatment. The BCG-treated 5367 and T24 cells showed higher levels of cytokine secretion than the cells in the 3D cell culture model (Table 1). In the 5637 cells, the secretion of IL-6 was increased by 13% in the untreated group in 2D-cultured cells. In T24 cells, IL-6 secretion was increased by 68% in the 2D cell environment and increased by 51% in the 3D cell environment compared to the control. In addition, the same results were obtained with IL-12 and IFN-γ. IL-12 secretion by BCG treatment in the 5637 cells was increased by 57% in the untreated group in the 2D cell environment and increased by 46% in the 3D cell environment. In T24 cells, there was an increase of 34% in the 2D environment

**Table 1. The concentrations of cytokines in 2D and 3D bladder cancer cell environments after BCG treatment.**

| | 5637 | | | | T24 | | | |
|---|---|---|---|---|---|---|---|---|
| | 2D | | 3D | | 2D | | 3D | |
| | untreated | BCG | untreated | BCG | untreated | BCG | untreated | BCG |
| **Cytokine level** | | | | | | | | |
| **(pg/mL, mean ± SD)** | | | | | | | | |
| IL-6 | 195.29 ± 2.30 | 220.66 ± 2.12** | 190.74 ± 2.42 | 198.45 ± 1.61 | 162.69 ± 4.81 | 274.31 ± 6.05** | 128.94 ± 4.66 | 195.21 ± 11.58* |
| IL-12 | 29.62 ± 0.66 | 46.75 ± 0.45** | 25.52 ± 0.99 | 37.44 ± 1.50* | 45.38 ± 1.36 | 60.86 ± 1.77* | 29.33 ± 0.44 | 34.65 ± 1.85 |
| INF-γ | 33.43 ± 1.00 | 40.69 ± 1.05** | 34.50 ± 0.54 | 36.57 ± 1.11 | 46.00 ± 0.23 | 51.54 ± 0.19** | 42.34 ± 0.97 | 44.92 ± 0.34 |

* $p < 0.05$ and

** $p < 0.01$

IL: interleukin, INF: interferon

compared with the untreated group. The secretion of IFN-γ increased by 21% in the 5637 cell 2D culture model. In T24 cells, the 2D cell culture model showed an increase of 12%. The secretion of cytokines was further activated with BCG in the 2D environment compared with the 3D environment (Table 1). In addition, there was a significant difference according to the amount of cytokine secretion in the 2D cell culture model, but there was no significant difference in the 3D cell culture model. Our results showed that when the effects of the drug were confirmed, cytokines secreted as a result of BCG treatment were increased in the 2D environment compared with those in the 3D environment.

## 6. The cell-to-cell interaction in 2D and 3D culture models

To measure E-cadherin and N-cadherin involved in the EMT mechanism, TGF-β1, an inducer of the EMT mechanism, treatment was applied (5 ng/ml) and measurement was performed using an ELISA kit. In the case of 5637 cells, E-cadherin (ng/ml, mean ± SE) showed a decrease of 4% in the treated group compared with the TGF-β1-untreated group in the 2D cell culture environment, but decreased by 17% in the TGF-β1-untreated group in the 3D cell culture environment. For the T24 cells in the 2D cell culture environment, the TGF-β1-treated group showed a decrease of 5% compared to the untreated group and decreased by 8% in the 3D cell culture environment. When the 2D cell culture model and the 3D cell culture model were compared to the control group, there was a significant difference in the amount of E-cadherin secreted by cells in the 3D cell culture model compared with those in the 2D cell culture model. In 5637 cells, N-cadherin showed an increase of 14% in the treated group compared with the TGF-β1-untreated group in the 2D cell culture environment, but increased by 45% in the 3D cell culture environment. For T24 cells in the 2D cell culture environment, the TGF-β1-treated group showed an increase of 7% compared with the untreated group and increased by 45% in the 3D cell culture environment. When the 2D cell culture model and the 3D cell culture model were compared to the control group, there was a significant difference in the amount of N-cadherin secreted in the 3D cell culture model compared with the 2D cell culture model. In addition, when the 2D cell culture model and 3D cell culture model were compared to the treatment group after treatment with TGF-β1 acting as an EMT stimulate, there was a significant difference in the secretion amount of N-cadherin (Fig 5).

## Discussion

Many researchers have studied 3D cell cultures in the form of spheroids using a matrix. Spheroid model has been made by various methods including spinner flask methods, gyratory

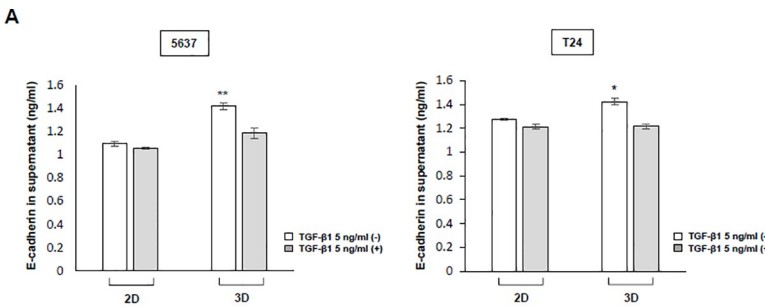

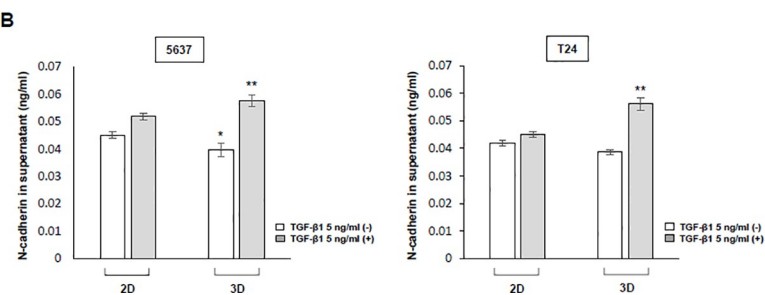

**Fig 5. Cell-to-cell marker protein expression in the 2D and 3D cell culture environment.** (A), (B) Induction of release of the epithelial-mesenchymal transition (EMT) mechanism protein in bladder cancer cells treated with TGF-β1. After 72 h, the supernatant of the 5637 and T24 bladder cancer cell culture media was separated, and e-cadherin and n-cadherin were measured by sandwich ELISA. * $p < 0.05$, ** $p < 0.01$, ratios were compared between the 2D- and 3D-cultured TGF-β1-untreated group and the 2D- and 3D-cultured TGF-β1-treated group. Data are the mean ± SEM of three independent experiments.

rotation systems, hanging drop cultures, surface-modified substrates or scaffolds, and micro-fabricated microstructures. In this study, we used dual extruder system (extruder & dispenser). The outer layer is mainly composed of viable cells, which tend to receive less oxygen, growth factors, and nutrients from the growth medium, and is in a stationary or low oxygen state.[14, 15] However, this model lacks ECM, an essential component of tumor biology. To overcome this, a new class of 3D bioprinting technology has been developed that enables the printing of hydrogel bio inks directly onto hydrogels, allowing fabrication of discrete patterns in the volumetric space over relatively large scales. By simple manipulation, the pattern engraved in the hydrogel block can be sacrificially removed to leave the desired cavity in the 3D space, or the support matrix can be washed to leave only the bioprinted 3D structure. Traditionally described as tools made of polymeric biomaterials, 3D scaffolds have the advantage of providing structural support for cell attachment and tissue development. Scaffolds provide a site of attachment that allows the cell's extracellular environment, ECM, to be reproduced, and the cell's ability to grow in 3D form, some of which are the stiffness and associated soluble factors such as growth factor or immune system of this environment. Using the 3D cell culture scaffold can facilitate oxygen, nutriment and waste transportation. Thus, cells can proliferate and migrate within the scaffold to eventually adhere on.[2]

In addition 3D cell culture that use scaffolds offer wanted size surface and are generally larger than those not relying on scaffold.[16]. The nature of the scaffold plays a decisive role in the presentation of substituents and the nature of the linkers linked to them. Several substituents, including Calix [n] arene or other polymers, have been used successfully as scaffolds for the multivalent presentation of sugar moieties. Calix [n] arenes are widely known for their

ability to interact with a wide range of biomolecules, resulting in a wide range of biological applications. For example, cationic calix [n] arene derivatives have been reported to be efficient DNA binders. [17] However, such derivatives pose some disadvantages in biology, including toxicity due to destabilization of cell membranes with negative potentials or nonspecific electrostatic aggregation by negatively charged DNA. GelMA, by contrast, is one of the most versatile hydrogels available for 3D cell culture and bioprinting with excellent biocompatibility, degradability and low cost. Therefore, we designed a cancer cell tissue model scaffold using GelMA with a 3D bio printer.

Our study found that longer UV exposure times significantly increase stiffness. This was consistent with previous studies that showed effects of UV on structure, pore size, and cell-to-cell interaction of structures.[18] According to another study, the suggested wavelength of UV is 250–400 nm [19]; the height is 50, 100, or 150 μM [20]; and the crosslinking time is 0–180 sec. [21]. We established an ideal crosslinking condition through experimentation by making a scaffold with GelMA at a height of 0.08 mm and then binding it with UV light at 356 nm for 120 sec. There may be a concern about DNA damage caused by UV exposure. But it is reported that DNA double-strand breaks were observed between 2 and 8 h after UV treatment, possibly resulting from replication fork collapse at damaged DNA sites. [22] Therefore, short time exposure of UV will not be a problem at all. In fact, in our experiments, we tested various settings of UV exposure time to minimize excessive UV exposure, and we comfirmed that there was no problem of cell survival following UV exposure by cell viability analysis.

Cells grown in 2D culture conditions are flat, stretched, and exhibit different gene and protein expression from those observed *in vivo*.[23] In 3D culture, cell-to-cell interactions are similar to those that occur *in vivo*, and there is a difference in the rate at which cells multiply.[24] When 5637 and T24 cells were cultured in 2D and 3D, fewer dead cells were found in 2D cell culture than in 3D cell culture. Also, when bladder cancer cells were cultured in 3D, they were found to have higher cell proliferation rates. Previous studies also suggested that cells grow and multiply rapidly because they mimic cell-to-cell interaction well in the 3D model.[14, 25] Additionally, compared with a previous study[26], there was a difference in the rate of reproduction across cell lines, and we found that 5637 and T24 cells proliferate faster in 3D cell culture (Fig 3B). For this reason, the 3D cell culture model provides a more realistic tumor microenvironment.

When 2D and 3D cell culture models were treated with drugs, the 2D-cultured cells showed reduced phosphorylation of mTOR and phosphorylation of sub-pathways compared with the 3D cultures. The mTOR controls the protein translations by phosphating the effector of 4E-BP1 and p70s6k by rapamycin.[27] However, this reaction is controlled in response to various stimuli, such as inducing an autophage from cancer cells, so it is important to check the effect rapamycin treatment[28], which is commonly used as a cancer cell inhibitor. Our results showed that there was a difference in viability between 2D and 3D cell cultures due to rapamycin. This was attributed to a difference in inhibition of mTOR phosphorylation. Therefore, our results confirmed that the 3D cell model was less sensitive to the drug, and this resistance was consistent with that observed in the previous studies[5, 14, 26]. We tried to compare 2D- and 3D-cultured cells by examining the immunoreactivity of BCG, one of the most effective and commonly used immunotherapies for bladder cancer patients. The study was the first report to identify a cytokine in a scaffold model created by cultivating bladder cancer cells in 3D. In previous studies[29], when we used BCG to identify the secretion of cytokines produced by BCG, as predicted, the amount of cytokines produced in the 3D cell culture model was less than that produced in the 2D cell culture model. However, a study should compare the effects of drugs in 2D and 3D cell cultures by confirming the amount of cytokines secreted through internalization directly affecting the anticancer effect. Based on these results, the drug effect in

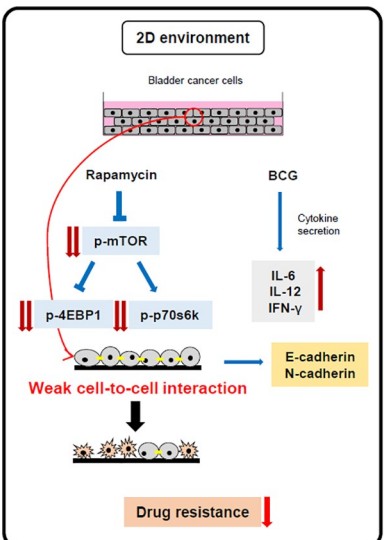
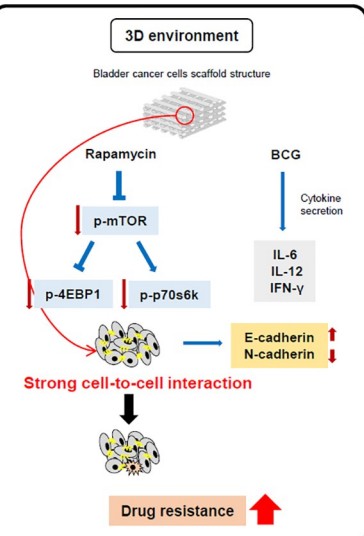

**Fig 6. Hypothetical schema of comparison of drug resistance effect in 2D and 3D environments due to cell-to-cell interaction.** The 2D cell culture environment and 3D cell culture environment reacted to rapamycin and BCG treatment, and there were strong differences. The mTOR signal pathway was more highly blocked by rapamycin in 2D than in 3D. The secretion of IL-6, IL-12, and IFN-γ, which are antitumor cytokines, due to BCG was also increased more in 2D than in 3D, resulting in an enhanced antitumor effect on bladder cancer cells. This indicates that our model confirms the secretion of E-cadherin and N-cadherin and shows the difference in drug resistance according to the difference in the intensity of cell-to-cell interaction. The results provide a rationale for evaluating drugs using the 3D cell culture environment.

the 2D cell culture model is exaggerated. This explains why rapamycin and BCG have shown excellent efficacy in research studies but not in clinical studies and patients.

EMT is a process in which epithelial cells lose cell-to-cell interactions and become mobile and invasive.[30, 31] Differences in growth in cell lines and drug effects can occur as a result of cell-to-cell interactions.[32] The 3D cancer models of *in vivo* cell-to-cell interactions are becoming important for drug testing and tumor biological studies.[33] The cadherin expressed in the EMT process is present in the membrane of the cell, and when the cadherin present in the cell membranes of two different cells binds, it induces a cell-to-cell interaction.[34] E-cadherin plays an important role in cell adhesion and is most expressed in association with cell-to-cell interaction.[35] Conversely, N-cadherin was expressed in the migration of cells through the progression of EMT and is expressed when the cells are disassociated and metastasized. [36] Since E-cadherin and N-cadherin are key markers for cell to cell interaction, our results show the difference in the secretion in 2D and 3D cell culture models. We expected that E-cadherin, a marker for cell-to-cell interaction, would be higher in the 3D cell culture model than in the 2D cell culture model. We measured E-cadherin in 5637 and T24 bladder cancer cells, and it was higher in the 3D cell model than in the 2D cell model, consistent with our expectations. Meanwhile, TGF-β1 acts as a strong factor in inducing EMT and promoting cancer.[37, 38] As the EMT progresses, E-cadherin is suppressed either directly or indirectly.[39] We expected to observe a higher N-cadherin concentration in 3D when cell-to-cell interaction induces TGF-β1 that induces EMT. However, the difference between E-cadherin and N-cadherin before and after treatment with TGF-β1 was not significant in the 2D cell culture model. However, the 3D cell culture model showed a difference in the amount of each cadherin after treatment with TGF-β1 because of the strong cell-to-cell interaction. This result supports our findings because the sensitivity of the drug can be changed as a result of cell-to-cell

interactions, and there was a difference in the permeability and effectiveness of the drug.[40] Therefore, the results suggest that the development of 3D cancer cell models that constitute a mechanism for cell-to-cell interaction may provide a better platform for testing drugs. Furthermore, we can use the 3D cell culture model to produce more clinically relevant results (Fig 6).

Our results showed that: first, the 3D bladder cancer cell culture model is faster to establish and more stable than the 2D bladder cancer cell culture model. Second, the 3D-cultured cancers cells show higher drug resistance and less sensitivity than the 2D-cultured cancer cells. Finally, the 3D cancer cell culture model shows cell-to-cell interaction and basal action that is similar to that observed in the *in vivo* environment. The 3D cancer cell culture model is essential for establishing a model similar to the patient cancer model. Its use will allow us to obtain more accurate results than those obtained using the 2D cell culture model for evaluating drug responses in the future. In addition, further research on organs-on-chips systems built by incorporating human-supplied tissues will develop personalized medicines for more accurate predictions of specific personal responses to drugs, thereby improving the treatment of cancer and other diseases.

## Acknowledgments

This research was supported by the Basic Science Research Program through the National Research Foundation of Korea (NRF) funded by the Ministry of Education, Science, and Technology, Republic of Korea (2017R1D1A1B03031514, 2018R1D1A1A02050248), and the Korea Health Technology R&D Project (HI17C0710). The funders had no role in study design, data collection and analysis, decision to publish, or preparation of the manuscript.

## Author Contributions

**Conceptualization:** Byung Hoon Chi, Young Mi Whang.

**Data curation:** Myeong Joo Kim.

**Formal analysis:** In Ho Chang.

**Investigation:** Young Mi Whang.

**Supervision:** In Ho Chang.

**Validation:** Byung Hoon Chi, James J. Yoo, Young Min Ju, Young Mi Whang, In Ho Chang.

**Writing – original draft:** Myeong Joo Kim.

**Writing – review & editing:** Myeong Joo Kim, James J. Yoo, Young Min Ju, In Ho Chang.

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
