## [Decision Letter · Decision Letter 0]

4 Jul 2019

PONE-D-19-14665

Establishment of a printed three-dimensional (3D) cell culture model for bladder cancer

PLOS ONE

Dear M.D. Chang,

Thank you for submitting your manuscript to PLOS ONE. After careful consideration, we feel that it has merit but does not fully meet PLOS ONE’s publication criteria as it currently stands. Therefore, we invite you to submit a revised version of the manuscript that addresses the points raised during the review process.

Two experts have reviewed the anuscript and found that the study was interesting and well described but still certain aspects of the current manuscript should be revised for a publication in the journal.

We would appreciate receiving your revised manuscript by Aug 18 2019 11:59PM. To enhance the reproducibility of your results, we recommend that if applicable you deposit your laboratory protocols in protocols.io, where a protocol can be assigned its own identifier (DOI) such that it can be cited independently in the future. For instructions see: http://journals.plos.org/plosone/s/submission-guidelines#loc-laboratory-protocols

We look forward to receiving your revised manuscript.

Kind regards,

Jung Weon Lee, Ph.D.

Academic Editor

PLOS ONE

Journal Requirements:

This research was supported by the Basic Science Research Program through the National Research Foundation of Korea (NRF) funded by the Ministry of Education, Science, and Technology, Republic of Korea (2017R1D1A1B03031514, 2018R1D1A1A02050248), and the Korea Health Technology R&D Project (HI17C0710).

Please remove any funding-related text from the manuscript and let us know how you would like to update your Funding Statement. Currently, your Funding Statement reads as follows: "No".

Reviewers' comments:

Reviewer's Responses to Questions

**Comments to the Author**

1. Is the manuscript technically sound, and do the data support the conclusions?

Reviewer #1: Yes

Reviewer #2: Yes

2. Has the statistical analysis been performed appropriately and rigorously? 

Reviewer #1: Yes

Reviewer #2: Yes

3. Have the authors made all data underlying the findings in their manuscript fully available?

Reviewer #1: Yes

Reviewer #2: Yes

4. Is the manuscript presented in an intelligible fashion and written in standard English?

Reviewer #1: Yes

Reviewer #2: Yes

5. Review Comments to the Author

Reviewer #1: This manuscript describes a tissue culture system based on bioprinting. A substantial amount of work has gone into the manuscript and the data is of high quality.

Points of criticism:

(1) The Introduction section describes the advantage of 3D cultures over 2D monolayer cultures and everyone will agree. It is not clear to me what sort of advantage 3D bioprinting provides over spheroids.

(2) Is the UV cross linking procedure expected to affect the cells?

(3) I don't quite see how one can make statements on "3D" cultures without showing the 3D structure of the bioprinted material. Have the authors made histological sections? I don't see that in the manuscript (if it is there and I missed it, the authors should put more emphasis on this).

(4) BCG is used here. Could the authors explain the use of immune therapeutics in a culture system devoid of immune cells?

Minor point:

the manuscript is difficult to read due to the presence of multiple figures in the text (section 4 and 5). The data are presented in the figures and there is really no need to repeat the actual figures in the text.

Reviewer #2: This paper by Inho Chang and coworkers describes a nice method and an interesting use of 3D cell scaffolds using GelMA: The data are interesting and the paper well written, showing that the effects of drugs in 2D are more exaggerated than those in 3D.

I think the authors should discuss whether there is space for orienting chemical systems, such as dendrimers or calixarenes for instance, to orient the organization of cells in 3D and help the 3D printing.

As an example of these systems, the authors should cite: 10.1021/acs.joc.5b00878; LEGNANI L., COMPOSTELLA F., SANSONE F., TOMA. Cone calix[4]arenes with orientable glycosylthioureido groups at the upper rim: an in-depth analysis of their symmetry properties. J. Org. Chem. 80, 7412-7418 (2015).

6. PLOS authors have the option to publish the peer review history of their article (what does this mean?). If published, this will include your full peer review and any attached files.

Reviewer #1: No

Reviewer #2: No

---

## [Author Response · Author response to Decision Letter 0]

24 Aug 2019

PONE-D-19-14665

Title : Structure establishment of three-dimensional (3D) cell culture printing model for bladder cancer

Reviewer #1: This manuscript describes a tissue culture system based on bioprinting. A substantial amount of work has gone into the manuscript and the data is of high quality.

Points of criticism:

(1) The Introduction section describes the advantage of 3D cultures over 2D monolayer cultures and everyone will agree. It is not clear to me what sort of advantage 3D bioprinting provides over spheroids.

Thank you for your indication. We agreed with your opinion and we added some explanation in introduction paragraph line 62-73 and in discussion paragraph line 280-297, line 387-390 as below

Introduction paragraph line 62-73

3D in vitro tumor models have been successfully used to evaluate efficacy and tissue pharmacokinetics of anticancer drugs. 3D spheroids models have been studied to reproduce the spatial organization and microenvironmental factors of in vivo micro-tumors more accurately, such as relevant gradients of nutrients and other molecular agents, and It is possible to generate cell-to-cell and cell-to-matrix interactions by them. Although more advanced compared to two-dimensional culture, 3D spheroid models lack major ECM elements of the tumor microenvironment. To overcome this, 3D bioprinting techniques with scaffold bioink made up of cellular material and additives such as growth factors, signaling molecules, etc. have been utilized. Compared to traditional tissue engineering methods, the technologies utilized by 3D bioprinting systems allow for greater precision in the spatial relationship between the individual elements of the desired tissue. As advances of computer aided design (CAD), 3D bioprinting offers great potential for regenerative medicine applications.

Discussion paragraph line 280-297

Many researchers have studied 3D cell cultures in the form of spheroids using a matrix. Spheroid model has been made by various methods including spinner flask methods, gyratory rotation systems, hanging drop cultures, surface-modified substrates or scaffolds, and micro-fabricated microstructures. In this study, we used dual extruder system (extruder & dispenser). The outer layer is mainly composed of viable cells, which tend to receive less oxygen, growth factors, and nutrients from the growth medium, and is in a stationary or low oxygen state. However, this model lacks ECM, an essential component of tumor biology. To overcome this, a new class of 3D bioprinting technology has been developed that enables the printing of hydrogel bio inks directly onto hydrogels, allowing fabrication of discrete patterns in the volumetric space over relatively large scales. By simple manipulation, the pattern engraved in the hydrogel block can be sacrificially removed to leave the desired cavity in the 3D space, or the support matrix can be washed to leave only the bioprinted 3D structure. Traditionally described as tools made of polymeric biomaterials, 3D scaffolds have the advantage of providing structural support for cell attachment and tissue development. Scaffolds provide a site of attachment that allows the cell's extracellular environment, ECM, to be reproduced, and the cell's ability to grow in 3D form, some of which are the stiffness and associated soluble factors such as growth factor or immune system of this environment. Using the 3D cell culture scaffold can facilitate oxygen, nutriment and waste transportation. Thus, cells can proliferate and migrate within the scaffold to eventually adhere on. 

Discussion paragraph line 387-390

In addition, further research on organs-on-chips systems built by incorporating human-supplied tissues will develop personalized medicines for more accurate predictions of specific personal responses to drugs, thereby improving the treatment of cancer and other diseases.

(2) Is the UV cross linking procedure expected to affect the cells?

Thank you for your indication. As you point out, there may be a concern about DNA damage caused by UV exposure. We added some explanation in discuss paragraph in line 316-322 as below

There may be a concern about DNA damage caused by UV exposure. But it is reported that DNA double-strand breaks were observed between 2 and 8 h after UV treatment, possibly resulting from replication fork collapse at damaged DNA sites. Therefore, short time exposure of UV will not be a problem at all. In fact, in our experiments, we tested various settings of UV exposure time to minimize excessive UV exposure, and we comfirmed that there was no problem of cell survival following UV exposure by cell viability analysis.

(3) I don't quite see how one can make statements on "3D" cultures without showing the 3D structure of the bioprinted material. Have the authors made histological sections? I don't see that in the manuscript (if it is there and I missed it, the authors should put more emphasis on this).

Thank you for your statement. We already confirmed the 3D structure of cultured materials by confocal microscope as provided in figure 2E. According to your opinion, we added some emphasis on result paragraph in line 196-199 as below

This can be controlled by stacking layers with many cells into 3D. Our ultimate goal is to maintain the structure of bladder cancer cells in 3D for more than 2D. So we confirmed that the shape remained after 5 days by comparing the GelMA only and cell/GelMA mixture structures (Fig. 2E).

(4) BCG is used here. Could the authors explain the use of immune therapeutics in a culture system devoid of immune cells?

Thank you for your indication. We have been doing some studies on antitumor effect of BCG related with autophagy of rapamycin. [1-5]

Despite long clinical experience with BCG, the mechanism of its therapeutic effect has been under investigation, but It is now widely accepted that two aspects seem to be important for effective antitumor effect. First, BCG-induced antitumor effects depend on a sequence of events involving attachment and internalization of BCG, secretion of cytokines and chemokines, and presentation of BCG and/or cancer cell antigens to cell of the immune system. Second, a direct antiproliferative, cytotoxic or proapoptotic effect of BCG on tumor cells is now also suggested. 

We conducted this study to clinically validate these findings. By measuring the levels of such cytokines in the bladder cancer cell models after BCG treatment, we think it could be possible to indirectly compare the therapeutic effect of BCG in each model. 

Minor point:

the manuscript is difficult to read due to the presence of multiple figures in the text (section 4 and 5). The data are presented in the figures and there is really no need to repeat the actual figures in the text.

Thank you for your indication. We agreed with your opinion and we corrected the result paragraph as below 

4. The effect of rapamycin and BCG in 2D and 3D culture models

Cells were evaluated using the CCK-8 assay. The cell viability of 5637 and T24 cells was inhibited by rapamycin and BCG treatment. However, the 2D cell culture showed a higher level of suppression than the 3D cell culture model. In the case of 5637 cells, the 2D cell culture showed a great decrease in cell viability in the control with each day in the untreated group due to rapamycin (1 μg/ml), and based on in the control, each untreated group in the 3D cell culture model was decreased. When Bacillus Calmette-Guérin (BCG) (30 MOI) was administered in each comparative control group, viability was decreased in the 2D and 3D cell culture model. T24 cells were also dramatically decreased by the rapamycin in 2D cell culture and this was also observed in the 3D cell culture model. When BCG was administered in the control group, viability decreased in the 2D cell culture environment and decreased in the 3D cell culture environment (Fig. 4A).

5. The BCG effect on cytokine production in the 2D and 3D cell culture environment

We measured the levels of several cytokines (IL-6, IL-12, and IFN-γ) in the 2D and 3D bladder cancer cell models after BCG treatment. The BCG-treated 5367 and T24 cells showed higher levels of cytokine secretion than the cells in the 3D cell culture model (Table 1). In the 5637 cells, the secretion of IL-6 was increased by 13% in the untreated group in 2D-cultured cells. In T24 cells, IL-6 secretion was increased by 68% in the 2D cell environment and increased by 51% in the 3D cell environment compared to the control. In addition, the same results were obtained with IL-12 and IFN-γ. IL-12 secretion by BCG treatment in the 5637 cells was increased by 57% in the untreated group in the 2D cell environment and increased by 46% in the 3D cell environment. In T24 cells, there was an increase of 34% in the 2D environment compared with the untreated group. The secretion of IFN-γ increased by 21% in the 5637 cell 2D culture model. In T24 cells, the 2D cell culture model showed an increase of 12%. The secretion of cytokines was further activated with BCG in the 2D environment compared with the 3D environment (Table 1). In addition, there was a significant difference according to the amount of cytokine secretion in the 2D cell culture model, but there was no significant difference in the 3D cell culture model. Our results showed that when the effects of the drug were confirmed, cytokines secreted as a result of BCG treatment were increased in the 2D environment compared with those in the 3D environment.

6. The cell-to-cell interaction in 2D and 3D culture models

To measure E-cadherin and N-cadherin involved in the EMT mechanism, TGF-β1, an inducer of the EMT mechanism, treatment was applied (5 ng/ml) and measurement was performed using an ELISA kit. In the case of 5637 cells, E-cadherin (ng/ml, mean ± SE) showed a decrease of 4% in the treated group compared with the TGF-β1-untreated group in the 2D cell culture environment, but decreased by 17% in the TGF-β1-untreated group in the 3D cell culture environment. For the T24 cells in the 2D cell culture environment, the TGF-β1-treated group showed a decrease of 5% compared to the untreated group and decreased by 8% in the 3D cell culture environment. When the 2D cell culture model and the 3D cell culture model were compared to the control group, there was a significant difference in the amount of E-cadherin secreted by cells in the 3D cell culture model compared with those in the 2D cell culture model. In 5637 cells, N-cadherin showed an increase of 14% in the treated group compared with the TGF-β1-untreated group in the 2D cell culture environment, but increased by 45% in the 3D cell culture environment. For T24 cells in the 2D cell culture environment, the TGF-β1-treated group showed an increase of 7% compared with the untreated group and increased by 45% in the 3D cell culture environment. When the 2D cell culture model and the 3D cell culture model were compared to the control group, there was a significant difference in the amount of N-cadherin secreted in the 3D cell culture model compared with the 2D cell culture model. In addition, when the 2D cell culture model and 3D cell culture model were compared to the treatment group after treatment with TGF-β1 acting as an EMT stimulate, there was a significant difference in the secretion amount of N-cadherin (Fig. 5).

Reviewer #2: This paper by Inho Chang and coworkers describes a nice method and an interesting use of 3D cell scaffolds using GelMA: The data are interesting and the paper well written, showing that the effects of drugs in 2D are more exaggerated than those in 3D.

I think the authors should discuss whether there is space for orienting chemical systems, such as dendrimers or calixarenes for instance, to orient the organization of cells in 3D and help the 3D printing.

As an example of these systems, the authors should cite: 10.1021/acs.joc.5b00878; LEGNANI L., COMPOSTELLA F., SANSONE F., TOMA. Cone calix[4]arenes with orientable glycosylthioureido groups at the upper rim: an in-depth analysis of their symmetry properties. J. Org. Chem. 80, 7412-7418 (2015).

Thank you for your kind comment. We agree with your opinion, so we corrected discussion paragraph in line 298-309 as below

In addition 3D cell culture that use scaffolds offer wanted size surface and are generally larger than those not relying on scaffold. The nature of the scaffold plays a decisive role in the presentation of substituents and the nature of the linkers linked to them. Several substituents, including Calix [n] arene or other polymers, have been used successfully as scaffolds for the multivalent presentation of sugar moieties. calix [n] arenes are widely known for their ability to interact with a wide range of biomolecules, resulting in a wide range of biological applications. For example, cationic calix [n] arene derivatives have been reported to be efficient DNA binders. However, such derivatives pose some disadvantages in biology, including toxicity due to destabilization of cell membranes with negative potentials or nonspecific electrostatic aggregation by negatively charged DNA. GelMA, by contrast, is one of the most versatile hydrogels available for 3D cell culture and bioprinting with excellent biocompatibility, degradability and low cost. Therefore, we designed a cancer cell tissue model scaffold using GelMA with a 3D bio printer.

1. Cho, M.J., et al., The immunotherapeutic effects of recombinant Bacillus Calmette-Guerin resistant to antimicrobial peptides on bladder cancer cells. Biochem Biophys Res Commun, 2019. 509(1): p. 167-174.

2. Whang, Y.M., et al., MEK inhibition enhances efficacy of bacillus Calmette-Guerin on bladder cancer cells by reducing release of Toll-like receptor 2-activated antimicrobial peptides. Oncotarget, 2017. 8(32): p. 53168-53179.

3. Choi, S.Y., et al., Modulating the internalization of bacille Calmette-Guerin by cathelicidin in bladder cancer cells. Urology, 2015. 85(4): p. 964 e7-964 e12.

4. Kwon, J.K., et al., Murine beta-defensin-2 may regulate the effect of bacillus Calmette-Guerin (BCG) in normal mouse bladder. Urol Oncol, 2015. 33(3): p. 111 e9-16.

5. Kim, J.H., et al., Human beta-defensin 2 may inhibit internalisation of bacillus Calmette-Guerin (BCG) in bladder cancer cells. BJU Int, 2013. 112(6): p. 781-90.

---

## [Decision Letter · Decision Letter 1]

26 Sep 2019

Structure establishment of three-dimensional (3D) cell culture printing model for bladder cancer

PONE-D-19-14665R1

Dear Dr. Chang,

We are pleased to inform you that your manuscript has been judged scientifically suitable for publication and will be formally accepted for publication once it complies with all outstanding technical requirements.

With kind regards,

Jung Weon Lee, Ph.D.

Academic Editor

PLOS ONE

Additional Editor Comments (optional):

Reviewers' comments:

Reviewer's Responses to Questions

**Comments to the Author**

1. If the authors have adequately addressed your comments raised in a previous round of review and you feel that this manuscript is now acceptable for publication, you may indicate that here to bypass the “Comments to the Author” section, enter your conflict of interest statement in the “Confidential to Editor” section, and submit your "Accept" recommendation.

Reviewer #1: All comments have been addressed

2. Is the manuscript technically sound, and do the data support the conclusions?

Reviewer #1: Yes

3. Has the statistical analysis been performed appropriately and rigorously? 

Reviewer #1: Yes

4. Have the authors made all data underlying the findings in their manuscript fully available?

Reviewer #1: Yes

5. Is the manuscript presented in an intelligible fashion and written in standard English?

Reviewer #1: Yes

6. Review Comments to the Author

Reviewer #1: (No Response)

7. PLOS authors have the option to publish the peer review history of their article (what does this mean?). If published, this will include your full peer review and any attached files.

Reviewer #1: No

---

## [Editor Report · Acceptance letter]

7 Oct 2019

PONE-D-19-14665R1 

Structure establishment of three-dimensional (3D) cell culture printing model for bladder cancer 

Dear Dr. Chang:

I am pleased to inform you that your manuscript has been deemed suitable for publication in PLOS ONE. Congratulations! Your manuscript is now with our production department. 

With kind regards,

on behalf of

Dr. Jung Weon Lee 

Academic Editor

PLOS ONE